# Open-Air Testing of Dual-Comb Time-of-Flight Measurement

**DOI:** 10.3390/s23218949

**Published:** 2023-11-03

**Authors:** Wooram Kim, Jaewon Yang, Jaeyoung Jang, Jeong Seok Oh, Seongheum Han, Seungman Kim, Heesuk Jang, Young-Jin Kim, Seung-Woo Kim

**Affiliations:** 1Department of Mechanical Engineering, Korea Advanced Institute of Science and Technology (KAIST), Yuseong-gu, Daejeon 34141, Republic of Korea; kwr0704@kaist.ac.kr (W.K.); obas@kaist.ac.kr (J.Y.); jayjang44@kaist.ac.kr (J.J.); janghsk@add.re.kr (H.J.); yj.kim@kaist.ac.kr (Y.-J.K.); 2Advanced Manufacturing Systems Research Division, Korea Institute of Machinery and Materials (KIMM), Yuseong-gu, Daejeon 34103, Republic of Korea; ojs6114@kimm.re.kr (J.S.O.); sh_han@kimm.re.kr (S.H.); kimsm@kimm.re.kr (S.K.)

**Keywords:** long ranging, optical cross-correlation, femtosecond lasers

## Abstract

We configured a long-distance ranging apparatus to test the principle of dual-comb time-of-flight measurement using ultrashort lasers. Emphasis was given to the evaluation of open-air performance quantitatively in terms of the measurement resolution and stability. The test results revealed that our dual-comb asynchronous optical pulse sampling permits micrometer-resolved ranging with a repeatability of 2.05 μm over a 648 m distance in dry weather conditions. Further atmospheric effects were evaluated in three different weather conditions with corresponding Allan deviations. Finally, the capability of simultaneous determination of multiple targets was verified with the potential of advanced industrial applications, such as manufacturing, surveying, metrology, and geodesy.

## 1. Introduction

Precision positioning using light-based distance ranging is essential in various fields of science and engineering [1]. In particular, the ability to measure long distances with sub-millimeter resolutions or less is strongly demanded in the field of large-scale precision engineering, such as for next-generation space missions and in shipbuilding/aviation industries. Conventional time-of-flight measurement, widely performed through the use of intensity-modulated or pulsed lasers, is not capable of providing such high resolutions, mainly because of electronics’ speed of light-to-electron signal conversion not being fast enough [2,3]. In addition, phase-measuring laser interferometry commonly available with sub-wavelength resolutions is restricted in its measurable range, as the phase coherence of the laser source deteriorates severely with increasing distance [4]. Meanwhile, along with the optical frequency comb made available in recent years with the development of mode-locked ultrashort lasers, several new comb-based principles have been proposed to extend the measurable range with enhanced resolutions by advancing the principles of phase-measuring interferometry [5]. First, synthetic wavelength interferometry (SWI) creates artificial microwaves by beating separate comb modes to handle several meters with micrometer-level resolutions [6,7,8,9,10,11,12]. Second, multiple-wavelength interferometry (MWI) employs multiple comb modes simultaneously to deal with a distance of several meters with a fine resolution in the sub-wavelength regime [13,14,15,16,17,18,19]. Third, dispersive interferometry (DI) exploits the interferometric phase slope of comb modes over a wide optical bandwidth to achieve a nanometer-level resolution within a range of a few micrometers [20,21,22,23,24]. Other comb-based interferometric principles tested so far include dual-comb dispersive interferometry [25,26,27,28], optical sampling by cavity tuning (OSCAT) [29,30], and the frequency modulation of continuous waves (FMCW) [31,32]. Despite many potential scientific, engineering, and industrial applications, the novel comb-based interferometric principles require more testing to validate their merits, along with much extensive experimental data under varied measurement conditions.

When the area of comb-based time-of-flight measurement is concerned, incorporating a femtosecond laser with the method of balanced optical cross-correlation (BCC) permits time-of-flight measurement to be implemented with a pulse timing resolution of less than 1 fs [33], as well as nanometer-level repeatability over a few kilometers [34,35]. Nonetheless, this BCC approach inevitably suffers a dead zone due to the limitation in sweeping the repetition rate of an ultrashort laser. This dead-zone problem was later overcome by combining the BCC method of optical cross-correlation with dual-comb asynchronous optical sampling [36,37,38,39,40,41]. Employing two distinct combs with slightly different repetition rates enhances pulse-timing resolution by spreading the pulse-timing temporal scale from nanoseconds to milliseconds. Accordingly, this dual-comb time-of-flight measurement provided nanometer-level precision ranging up to several meters. In addition, measurable range confinement within a single pulse-to-pulse interval was removed by adopting the Vernier effect by varying the pulse repetition rate by cavity length control [10,28,29]. Further, required dual-comb setups were realized in various ways: through repetition rate scanning using a signal laser [36], triple comb using two signal lasers with different repetition rates [42], or role switching between the signal laser and the local oscillator [38]. These attempts to extend the measurable range beyond a single pulse-to-pulse interval provide their own distinctive merits and disadvantages in terms of the achievable minimum resolution, post-processing time for signal analysis, and system hardware complexity.

In the authors’ previous study of Ref. [43], an absolute ranging system was proposed to achieve sub-micrometer resolutions based on time-of-flight measurement by dual-comb optical cross-correlation. In this investigation, the earlier dual-comb system is extended with a measurable range to 12 km by combining two stabilized combs of different mode spacings. Then, a series of open-air tests is performed for a distance target of 648 m, demonstrating a repeatability of 2.05 μm to verify the micrometer-level capability of long-ranging. In addition, the effect of atmospheric noise due to weather conditions on the long-term repeatability of measured distances is discussed. Finally, an evaluation is made of the possibility of achieving a resolution of less than a micrometer, conducted through frequency analysis by modulating multiple targets with different frequencies

## 2. Dual-Comb Long-Distance Ranging by Optical Cross-Correlation

### 2.1. Measurement Principles

Figure 1 describes the dual-comb apparatus of time-of-flight measurement configured in this investigation. The light source, as shown in Figure 1a, consists of two separate ultrashort lasers; one is to produce the measurement pulses at a repetition rate of *f_r_* and the other emits the local oscillator pulses at *f_r_* + ∆*f_r_*, with ∆*f_r_* being a slight repetition rate difference. The returning measurement pulses are overlapped with the local oscillator pulses through an optical nonlinear crystal made of periodically poled potassium titanyl phosphate (PPKTP). This crystal creates a second-harmonic pulse every moment, in proportion to the overlapped pulse strength, when the measurement pulse meets the local oscillator pulse. This dual-comb method permits the asynchronous optical sampling of the measurement pulses at a prolonged period of *t_slip_* = ∆*f_r_*/*f_r_*^2^ with a scale factor of *M* = *f_r_*/∆*f_r_*, as depicted in Figure 1b. The timing resolution before the time scale expansion is not accurate enough due to the pulse broadening in the used photodiode having a slow response time. However, Figure 1c shows how the measurement pulse is reconstructed in the expanded time scale without pulse broadening. The time-of-flight of the measurement pulse is gauzed with respect to the reference pulse reflected from a reference mirror as shown in Figure 1d,e. This implies that even if the used photodetector suffers a slow response time, usually in the nanosecond time scale, the pulse-timing resolution can be better than 1 fs [40,43]. Therefore, the target distance *D* is determined as
(1)$$D=\frac{c}{2N}∆T_{t.o.f}=\frac{c}{2N}\frac{∆f_{r}}{f_{r}}∆\tau$$
where *D* is the distance measured between the reference mirror and target mirror, *c* is the speed of light in vacuum, *N* is the group refractive index of air, ∆*T_t.o.f_* is the time-of-flight in the actual time scale, and ∆*τ* is the time-of-flight measured in the expanded time scale.

The pulse timing in the expanded time scale can be obtained by measuring the representative timing of each pulse. Here, the pulse timing is defined as the location of maxima in the 2nd order polynomial fitting curve of the reconstructed pulse, as shown in Figure 1f,g. There are two steps for pulse timing determination; first, 2nd order polynomial fitting is performed using 10 points around the maximum value of the pulse signal. Next, the timing corresponding to the extreme value of the 2nd order polynomial fitting curve is obtained. Thus, the time-of-flight, defined as ∆*τ* = *t_mea_* − *t_ref_*, can be determined by determining the timing gap of two reconstructed pulses. As shown in Figure 1f, the 2nd order polynomial fitting curve (orange) is derived from the original reference pulse (violet), and then the pulse timing *t_ref_* is calculated through a 1st-order differentiation operation. The measurement pulse timing *t_mea_* can be obtained as shown in Figure 1g. As a result, the high resolution of better than 1 fs is achieved by optical cross-correlation combined with 2nd order polynomial fitting.

### 2.2. Testing Setup

Figure 2 shows the testing setup configured for long-distance measurement in the main campus of KAIST in Daejeon, between two buildings of N1 and N7. The absolute laser ranging apparatus was installed on the rooftop on N7 and a target mirror was set on the rooftop of N1, as shown in Figure 2a. The absolute laser ranging apparatus consists of the dual-comb light source, optical cross-correlator, and real-time data processor (Figure 1a).

The dual-comb light source was configured by combining two ultrashort lasers, referred to as Laser 1 and Laser 2, as illustrated in Figure 1a. Each laser is a linear-type oscillator made of an Er-doped gain fiber and a semiconductor saturable absorber mirror (SESAM) as a mode-locking device [44]. The repetition rate *f_r_* was set at 194.5 MHz, and the repetition rate difference ∆*f_r_* was set at 12.4 kHz, with stabilization to the Rb atomic clock with a phase-locked-loop bandwidth of 100 Hz. The average output power of each oscillator was ~1 mW, which was amplified to 100 mW using an Er-doped fiber amplifier (EDFA). The center wavelength is 1558 nm, and the spectral bandwidth is 7.4 nm for Laser 1 and 5 nm for Laser 2. The pulse duration is 240 fs for Laser 1 and 205 fs for Laser 2. The pulse width of Laser 1 was broadened to 1.7 ps for accurate detection of the pulse-to-pulse overlap (PPO) and the timing jitter buildup (TBJ) [43].

The two lasers go through a 2 × 2 optical switch with a 0.5 ms frame rate, with the measurement laser being assigned on channel 1 and the local oscillator laser on channel 2, as shown in Figure 1a. The measurement laser was then launched into free space through an optical circulator combined with a collimating lens, with its beam diameter was expanded from 2 mm to 40 mm through a beam expander. The pulse train of the measurement laser is sequentially reflected from a pellicle beam splitter as well as the target mirror installed at a remote site. Then, the reflected pulse train enters the optical cross-correlator and meets the local oscillator from channel 2. Finally, the two pulses are converted into second-harmonic pulses through the PPKTP crystal, from which the distance signal is acquired and converted to a distance value through the real-time data processor seen in Figure 1a.

### 2.3. Extension of Measuring Range

Let *D_NAR_* be the non-ambiguity range (NAR) that denotes the extent of distance measurable without ambiguity. In principle, *D_NAR_* corresponds to the pulse-to-pulse interval of the measurement laser, which is given as
(2)$$D_{NAR}=\frac{c}{2Nf_{r}}$$

The actual distance *D* to be determined is therefore expressed with the introduction of a multiple integer *m* as
(3)$$D=m\cdot D_{NAR}+D_{F}$$
with *D_F_*, the fractional distance, actually being measured to be shorter than *D_NAR_* from Equation (2). Here, the multiple integer *m* is decided by swapping the repetition rate of the measurement laser with that of the local oscillator laser through the optical switch, as depicted in Figure 1a. Specifically, two separate measurements are performed in sequence; one with *f_r_*_1_ and the other *f_r_*_2_ = *f_r_*_1_
*−* Δ*f_r_*, as illustrated in Figure 2b. With two consecutively measured values of *D_NAR_*_1_ and *D_NAR_*_2_ corresponding to *f_r_*_1_ and *f_r_*_2_, respectively, the distance *D* is determined as [38],
(4)$$D=m_{1}D_{NAR1}+D_{F1}=m_{2}D_{NAR2}+D_{F2}$$

Now, if *D_F_*_1_
*> D_F_*_2_, the distance *D* is given as *m*_1_ = *nint*[(*D_F_*_1_
*− D_F_*_2_)/(*D_NAR_*_2_
*− D_NAR_*_1_)] and *m*_1_ = *m*_2_. Note that the *nint*[.] means the nearest integer to the real value within the bracket. On the other hand, if *D_F_*_1_ < *D_F_*_2_, it is decided as *m*_1_= *nint*[(*D_NAR_*_1_ + *D_F_*_1_
*− D_F_*_2_)/(*D_NAR_*_2_
*− D_NAR_*_1_)] and *m*_1_ = *m*_2_ + 1. Therefore, when measuring a long distance using two repetition rates, the expanded measurable range is given as *D_NAR,max_* = *c*/(2*N*∆*f_r_*). Here, *m*_1_ and *m*_2_ are integers for each repetition rate, respectively, and *D_F_*_1_ and *D_F_*_2_ represent the fractional distances determined within *D_NAR_*_1_ and *D_NAR_*_1_, respectively.

### 2.4. Test Results

Figure 2 shows a test result of long-distance measurement using the measurement apparatus seen in Figure 1. As shown in Figure 2c, the optical switch changes the repetition rate of the signal laser from *f_r_*_1_ to *f_r_*_2_, with *f_r_*_1_ and *f_r_*_2_ being 194.496100 MHz and 194.483700 MHz, respectively. The repetition rate difference was set at 12.4 kHz. Then *D_NAR_*_1_ and *D_NAR_*_2_ were 770.690 mm and 770.739 mm, respectively, and the maximum measurable range without ambiguity was ~12 km. Figure 2d reveals that the fractional distances *D_F_*_1_ and *D_F_*_2_ vary from 559.376 mm to 518.106 mm, from which the measurement repeatability of single-shot measurement was calculated to be 7.84 um and 6.84 um, respectively. The integer *m* was determined, using the measurable range and fractional distance for each repetition rate, as 840. Therefore, through the correction of *m*, it was confirmed that the distance between the two buildings was 647.939 m. The measurement precision of the fractional distance influences the determination of the integer, which can be expressed in the form of
(5)$$m=nint\left[\frac{\left(D_{F1}+\delta D_{F1}\right)-\left(D_{F2}+\delta D_{F2}\right)}{D_{NAR2}-D_{NAR1}}\right]$$

Here, *δD_F_*_1_ and *δD_F_*_2_ represent the temporal fluctuations monitored by the moving averages of the measured values. If the measurement precision is low, the integer *m* is placed in the range of 840 ± 2. When the same experiment was repeated 1000 times, as shown in Figure 2e, the integer *m* is finalized to be 840 with a probability of 61.6%. Further, it is important to note that the measurement precision can be enhanced through time averaging with removal of the random noise causing timing jitter. For instance, the integer *m* can be made to converge to a single value at an averaging time of 80.65 ms (corresponding to 1000 points average). Figure 2f shows a measurement result for an interval of 80 s, in which the distance between N1 and N7 was decided to be 648 m with a single-shot repeatability of 7.76 μm, with an average time of 80.65 μs.

## 3. Performance Evaluation

### 3.1. Measurement Repeatability

Figure 3 presents the measured data for a fixed distance of 648 m taken in three different weather conditions: sunny, windy, and rainy. Through real-time signal processing, the same distance was repeatedly measured at a sampling rate of 9 ms during a total period of 3000 s for each weather condition. The time series plots of the measured distance in Figure 3a reveal that the measurement resolution lies in the micrometer regime, even though the measured data experience open-air temporal drift induced as the refractive index of air changes with various outdoor parameters such as temperature, pressure, and humidity. As far as the uncertainty of the measured data is concerned, it is presumed to lie in the range of ~1 × 10^−6^ in our measurements where no compensation was made for the refractive index of air. The amount of open-air drift is found to be within ~1 mm, equivalent to ~2 × 10^−7^ in fractional terms, not exceeding the estimated uncertainty level. Nonetheless, on the rainy and windy day, atmospheric turbulence causes short-term fluctuations, deteriorating the achievable measurement precision significantly.

For more systematic analysis, Figure 3b shows the measurement repeatability calculated in the form of Allan deviation. The influence of weather conditions is compared with respect to reference data (blue navy) obtained from a short distance of 3.0 m under a well-isolated environmental condition. Specifically, the reference data indicate that our dual-comb scheme is capable of providing sub-micrometer repeatability for indoor measurements for a distance of a few meters. On the other hand, the open-air data reveal far worsened repeatability; even in the sunny outdoor condition (orange), the repeatability is 7.38 μm for single-shot measurements, which improves to 2.05 μm when the averaging time increases to 1 s. As expected, the rainy and windy weather causes further deterioration, yielding no steady improvements for longer, even averaging at 1.0 s.

Figure 3c presents the power spectrum densities (PSDs) of phase noise caused by the effect of environmental noise on our measurement precision. In comparison to the case of indoor reference data of 3 m distance (blue navy), the sunny weather (orange) suffers phase noise caused by atmospheric turbulence exceeding 60 dB. Note that the *f^−^*^8/3^ noise line is clearly observed in the region of 10 mHz or less, and the *f^−^*^2^ noise line caused by random walk of the optical fiber of our interferometer setup are also seen in the region of 10 mHz to 100 mHz [45,46,47,48]. Further, the rainy and windy weather experiences a significant instability of the refractive index of air due to raindrops (water), revealing that the influence of atmospheric disturbance and refractive index change turns out to be the dominant limiting factor on the achievable measurement precision.

### 3.2. Dynamic Vibration Measurement

In addition to the measurement repeatability discussed in Figure 3, the capability of dynamic vibration measurement of our dual-comb scheme was evaluated via the experiment setup configured in Figure 4; the target mirror was modulated using a PZT actuator at the remote site. The modulation amplitude was fixed at 10 μm, while the modulation frequency was varied as 1, 5, 10, and 15 Hz. Figure 4b shows the measured data at each modulation frequency collected at a 9 ms sampling rate. Figure 4c presents the FFT plots of the measured amplitudes, confirming that initially there is no particular noise peak appearing when no modulation is given (grey). With modulation, the measured amplitude decreases with increasing the modulation frequency, from 5 μm at 1 Hz to 2.5 μm at 15 Hz. The reduction in the measured amplitude is attributed mainly to the mechanical inertia of the corner cube retroreflector used as the target mirror. It was consequently concluded that the mechanical vibration is determined with a 2.5 μm resolution up to 15 Hz, even at a distance of 648 m.

## 4. Key Advantages of Absolute Distance Ranging

### 4.1. Experimental Setup

Figure 5 shows another experiment result obtained to demonstrate the absolute measurement capabilities of our dual-comb time-of-flight scheme. Firstly, in contrast to widely used incremental laser interferometry, our absolute range is able to keep track of the target position even in the presence of measurement laser beam interruption. Specifically, with the experimental setup shown in Figure 5a, the signal recovery ability of our measurement was tested by blocking the measurement laser beam intentionally through the use of a mechanical chopper in rotation. The traced distance data of Figure 5b verifies that neither signal loss nor phase noise was introduced, with almost no delay in signal recovery. Secondly, our measurement scheme is capable of dealing with multiple targets simultaneously, which was verified with respect to the two distant target mirrors, M_1_ and M_2_, installed at the remote site on a linear stage with an offset distance of D_1_ − D_2_ = 17 mm.

### 4.2. Simultaneous Multi-Target Detection

Along the measurement beam line, our measurement system is able to deal with multiple targets simultaneously. With the expansion of the measurement beam to a 40 mm diameter, two target mirrors, M_1_ and M_2_, were installed at a distance of 648 m by sharing the measurement beam. Within the cross-section of the measurement beam, the target mirrors were not overlapped, with a separation gap of 40 mm and a offset distance of 75.645 mm. As shown in Figure 5c, the absolute distances for the two targets were measured to be 647.923 653 m and 647.999 299 m, respectively, for M_1_ and M_2_. The single-shot precision was 5.68 μm and 4.01 μm, respectively, with fluctuations of 482.253 mm and 560.899 mm. The target M_1_ showed relatively worse precision, as it happened to be close to D_NAR/2_ [39,43].

Now, in order to verify the possibility of the precise positioning of a moving target at the remote site, the target mirrors, M_1_ and M_2_, were located on a linear stage that was set to reciprocate with a stroke of 90 mm. A commercial linear encoder was also installed on the stage, of which the output signal was used as a reference. Figure 5d shows a measurement result; the orange graph represents the absolute distance value of M_1_ obtained from our measurement system, and the pink graph represents the encoder value acquired at the same time. The absolute distance of M_1_ varies from 647.865 m to 647.955 m by the motion of the stage, with a distance gap of 90 mm between M_1_ and the encoder. The result demonstrates the possibility of precise positioning of a moving target located at the remote site through our ranging system.

## 5. Conclusions

The method of dual-comb time-of-flight measurement has been tested in this study for long-distance ranging without ambiguity up to 12 km by repetition rate swapping between the signal laser and local oscillator. In terms of the Allan deviation, the measurement precision was found to be as good as 2.05 μm at 1 s averaging for a distance of 648 m in dry weather conditions. In addition, three different weather conditions were tested to investigate how the weather-dependent instability caused by air turbulence affects the measurement precision. Furthermore, the micrometer-level measurement resolution was also confirmed by modulating the target distance with an amplitude of several micrometers at a sequence of preassigned frequencies, and subsequently resolving the modulation amplitude clearly at all the modulation frequencies up to 15 Hz. This high-precision capability was also established for the task of simultaneously positioning two targets in movement at the remote site. With remarkable advantages not realizable with conventional laser ranging devices, this comb-based absolute ranging method is expected to be widely used in many novel applications, such as next-generation space missions, particularly for satellite formation flying and large-scale manufacturing, including aviation/ship building.

## Figures and Tables

**Figure 1 sensors-23-08949-f001:**
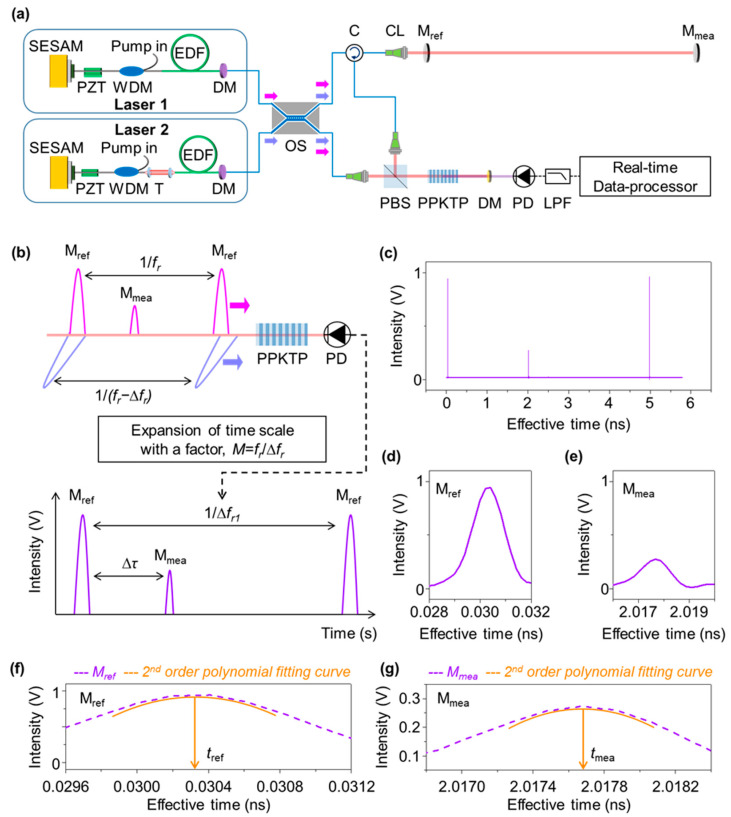
Dual-comb long-distance measurement. (**a**) Measurement system configuration. (**b**) Asynchronous optical sampling (ASOPS); pulse timing using second-harmonic generation. (**c**) Reconstructed pulse signals. (**d**) Reconstructed reference pulse. (**e**) Reconstructed measurement pulse; pulse timing in expanded time scale. (**f**) Reference pulse fitted with a 2nd order polynomial. (**g**) Fitting curve of the measurement pulse. SESAM: semiconductor saturable absorber mirror, PZT: piezoelectric transducer, WDM: wavelength division multiplexer, T: repetition rate tuner, EDF: erbium-doped fiber, DM: dichroic mirror, OS: 2 × 2 optical switch, C: optical circulator, CL: collimating lens, M_ref_: reference mirror, M_mea_: measurement mirror, PBS: polarization beam splitter, PPKTP: periodically poled KTP, PD: photodetector, LPF: low-pass filter.

**Figure 2 sensors-23-08949-f002:**
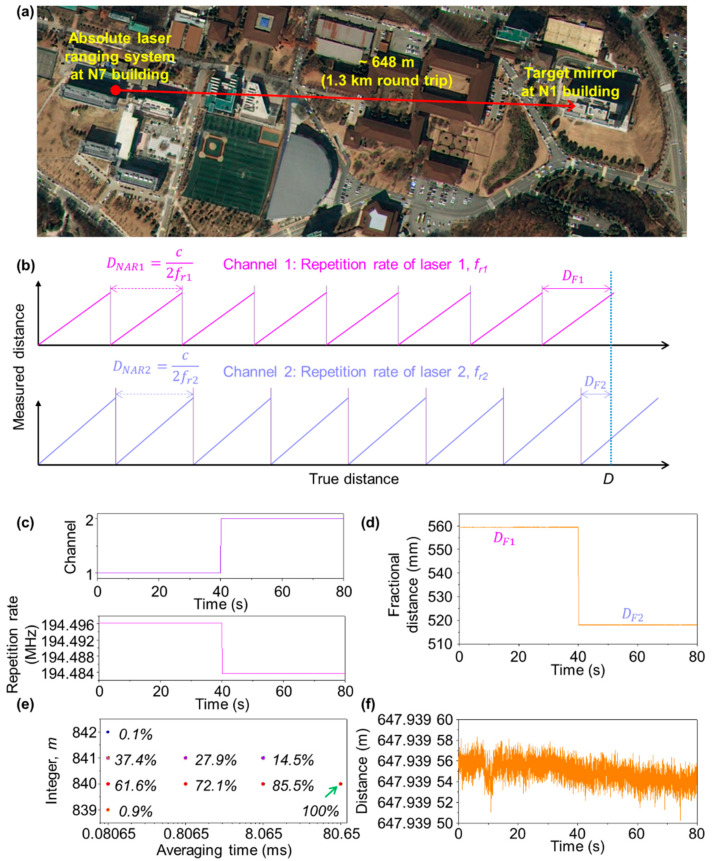
Experimental results. (**a**) Measurement of long distance between two buildings at KAIST. (**b**) Determination of fractional distances by repetition rate switching. (**c**) Repetition rate change with channel switching. (**d**) Fractional distance change. (**e**) Determination of the integer, *m*, depending on the precision measurement of the fractional distance. (**f**) Temporal fluctuation of long distance after correction of integer *m*.

**Figure 3 sensors-23-08949-f003:**
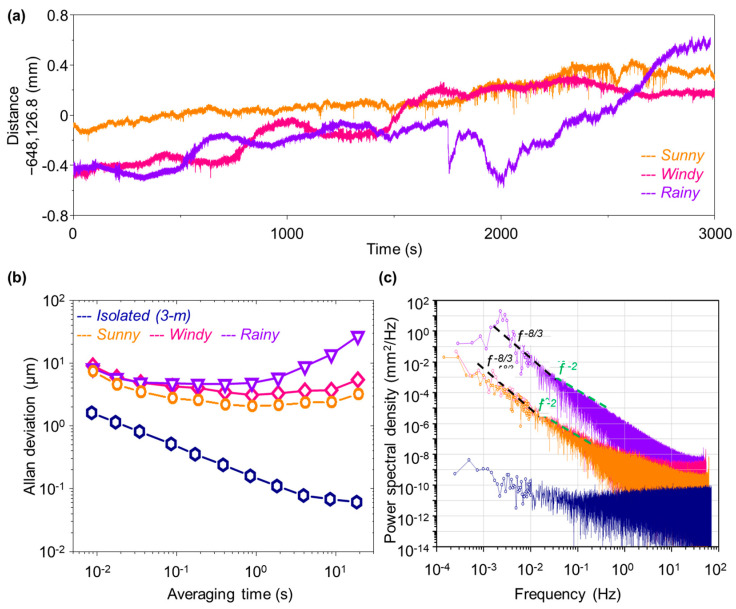
Measurement data for three different weather conditions for a representative distance of 648 m. (**a**) Temporal fluctuations of measured distance values over a period of 3000 s. (**b**) Measurement precision evaluated in terms of the Allan deviation. For comparison, a reference result obtained for a short distance of 3 m under a well-isolated laboratory environment is also shown. (**c**) Power spectral density of measured distance noises. Colors denote the same weather conditions as those of (**b**).

**Figure 4 sensors-23-08949-f004:**
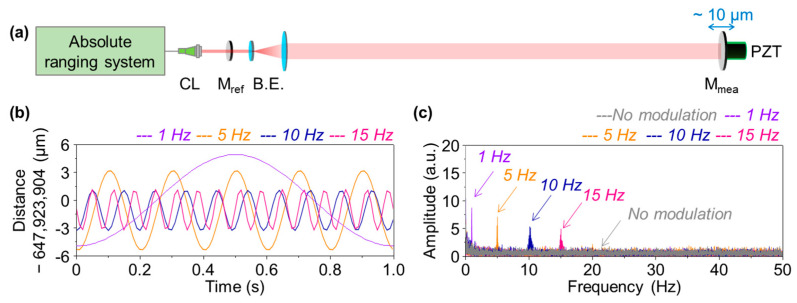
Experimental data for evaluation of measurement resolution. (**a**) Modulation of the target mirror using a PZT actuator. (**b**) Temporal variation in the measured distance at modulation frequencies of 1, 5, 10, and 15 Hz. (**c**) Amplitudes of the measured distance with varying modulation frequencies. CL: collimator, M_ref_: reference mirror, M_mea_: target mirror, and B.E.; beam expander.

**Figure 5 sensors-23-08949-f005:**
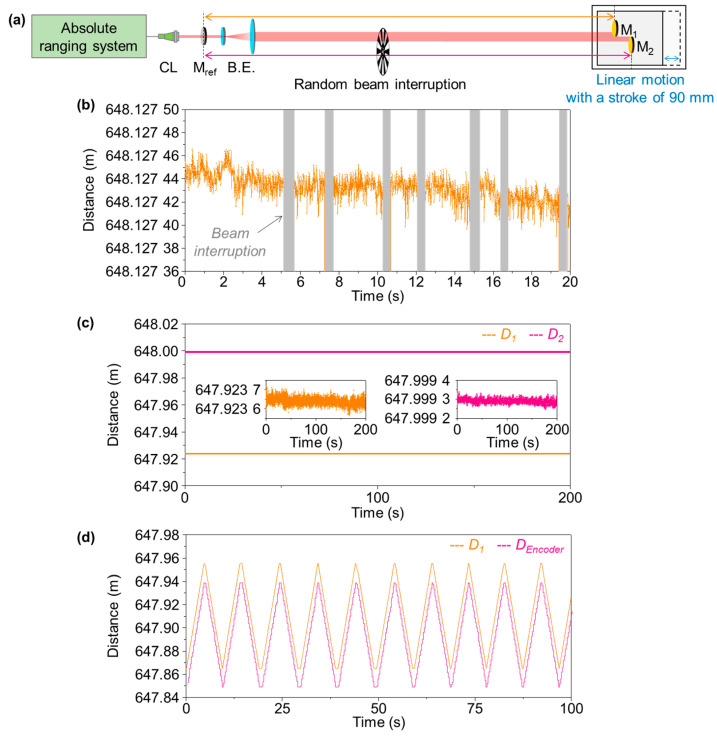
Absolute measurement of dual moving targets. (**a**) Experimental setup. (**b**) Absolute ranging with sudden beam interruption. (**c**) Simultaneous position determination of dual targets. (**d**) Comparison of absolute ranging to a tooth-saw stroke of 90 mm read using an optical encoder. CL: collimator, M_ref_: reference mirror, and B.E.; beam expander.

## Data Availability

Data supporting reported results can be found by contacting the authors.

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
