# Peer review of "Open-Air Testing of Dual-Comb Time-of-Flight Measurement"

_sensors, 2023, doi:10.3390/s23218949_

Round 1

Reviewer 1 Report

Comments and Suggestions for Authors

In the manuscript entitled “Open-air testing of dual-comb time-of-flight measurement,” the authors have proposed an absolute long-distance measuring method using two femtosecond lasers with different repetition rate. The method was capable to measure a distance of about 648 m with 2.05 micrometer precision in open-air with good weather condition. The developed apparatus has good potential for several applications and can be extended to measure up to 12 km without ambiguity. In my opinion, the manuscript can be accept for publication.

Author Response

The authors appreciate the reviewer’s comments.

Reviewer 2 Report

Comments and Suggestions for Authors

Very interesting results are presented in the paper. The relevance for applications of long-distance measurement is clear. Although not in the scope of this paper, these long-distance measurements will have to be complemented with methods able to include the effect of a varying refraction index over periods of weeks to have an "absolute" measurement.

l 285-287: "The 285 reduction of the measured amplitude is attributed mainly to the mechanical inertia of the corner-cube retroreflector used as the target mirror.": is the amplitude compatible with a f^-2 behaviour?

Minimal typographical errors have been spotted:

l 32: point instead of comma: [2,3], Besides -> [2,3]. Besides

l 221: there is a "be" missing

l 254: I would call it "blue navy"

l 315: one point too much: encoder. . CL -> encoder. CL

Author Response

Very interesting results are presented in the paper. The relevance for applications of long-distance measurement is clear. Although not in the scope of this paper, these long-distance measurements will have to be complemented with methods able to include the effect of a varying refraction index over periods of weeks to have an "absolute" measurement.

"The 285 reduction of the measured amplitude is attributed mainly to the mechanical inertia of the corner-cube retroreflector used as the target mirror.": is the amplitude compatible with a f^-2 behaviour?

Yes, the mechanical structure holding the corner cube can be regarded as a typical second-order vibration system. Thus, the measured amplitude shows a damped vibration behavior, decreasing with increasing the modulation frequency as observed in Fig. 4(c).

Minimal typographical errors have been spotted:

l 32: point instead of comma: [2,3], Besides -> [2,3]. Besides:

 Yes. the typo has been corrected as “Conventional time-of-flight measurement widely performed by use of intensity-modulated or pulsed lasers is not capable of providing such high resolutions mainly because of not-fast-enough electronics speed of light-to-electron signal conversion [2,3].”

l 221: there is a "be" missing:

 Yes. It has been corrected as “Further, it is important to note that the measurement precision can be enhanced by time averaging with removal of the random noise causing timing jitter.”

l 254: I would call it "blue navy"

 Yes. The suggestion has been reflected as “The influence of weather conditions is compared with respect to a reference data (blue navy) obtained from a short distance of 3.0 m under a well isolated environmental condition.”

“In comparison to the case of indoor reference data of 3-m distance (blue navy), the sunny weather (orange) suffers phase noise caused by atmospheric turbulence exceeding 60 dB.”

l 315: one point too much: encoder. . CL -> encoder. CL:

 Yes. the typo has been corrected as “(d) Comparison of absolute ranging to a tooth-saw stroke of 90 mm read using an optical encoder. CL: collimator, Mref: reference mirror, and B.E.; beam expander.”

Reviewer 3 Report

Comments and Suggestions for Authors

The paper is devoted to the investigation of a long-distance ranging apparatus based on the ultrashort lasers. The authors have obtained interesting experimental results with valuable applied significance.  

The obtained results are beyond the doubt and the presentation of the paper is of high quality. 

But there are several questions to improve the matter of paper:

- Is it necessary to stabilize optical combs for each laser? And how was it implemented?  

- To what extent is the developed method applicable to real conditions? How difficult is it to implement this method in a real hardware?

After answering the questions, the paper can be published. 

Author Response

- Is it necessary to stabilize optical combs for each laser? And how was it implemented?  

In order to determine the ambiguity integer m without error, the repetition rate of each comb has to be stabilized as shown in Fig. 2(e). For this, the well-established technique of phase-locked loop (PLL) control is employed as follows;

  1. The target repetition rate each comb is set in the first place.
  2. The actual repetition rate is measured using a frequency counter stabilized to the Rb clock.
  3. PLL control signal is generated to lock the actual repetition rate to the target value.
  4. Then PLL control is activated by adjusting the cavity length of each comb using a PZT actuator
  5. Finally, ensure that the PLL-stabilized repetition rate maintains a stability of 10-11 at 0.5 s.

- To what extent is the developed method applicable to real conditions? How difficult is it to implement this method in a real hardware?

This dual-comb measurement system is intended for actual industrial applications such as large-scale ship manufacturing, aircraft assembly, building safety diagnosis, and space missions. Main difficulties lie in securing a stable dual-comb system; firstly, two combs are needed with a tiny, but consistent, different repetition rate. Secondly, two combs need to be stabilized synchronously to determine the ambiguity integer m with no error, particularly for long distances. Thirdly, a steady beam pointing is essential to avoid atmospheric disturbance between the target mirror and source laser.
